# "This is an absolute witch-hunt for nursing, and nobody feels safe." A qualitative study of nurses, mental health, and the criminal prosecution of a nurse error

**Patricia Dekeseredy** [1]*, **Treah Haggerty**[2], **Cara L. Sedney**[1]

**1** Department of Neurosurgery, Rockefeller Neuroscience Institute, West Virginia University, Morgantown, West Virginia, United States of America, **2** Department of Family Medicine, West Virginia University, Morgantown, West Virginia, United States of America

\* patricia.dekeseredy@hsc.wvu.edu

## Abstract

In March 2022, nurse RaDonda Vaught made headlines after being found guilty of two felony charges resulting from a fatal medication error. Compounded by the ease and availability of social media and speedy internet access, information regarding the public prosecution of medical errors spreads quickly. The extensive coverage of this case polarized the nursing community, prompting online discussions and generating responses from regulatory bodies. Regulatory authorities predicted this intense publicity may influence nurses not to report medication errors that they suspect could jeopardize their employment, reputation, or nursing license. This study is one of few to quantify these concerns through qualitative inquiry. The analysis of semi-structured interviews with practicing bedside nurses reveals four main themes. These are how negative disciplinary actions lead to fear of reporting medical errors, how nurses are held responsible for system failures, how real-life errors are common and inevitable, and the broader negative impact of prosecuting medical errors. These findings support that criminal prosecution and ensuing media coverage of nurse errors can have a broader negative impact on healthcare and patient safety. The current study is one of few to quantify the concerns of regulatory bodies that the publicity surrounding the prosecution would negatively impact nurses' practice and workplace mental health. Considering the impact of intense media coverage, more work is indicated to identify safe and constructive ways to handle medical mistakes that do not perpetuate fear of reporting them. Since the criminal conviction in the RaDonda Vaught case in March 2024, Kentucky became the first state to address this by enacting protections for its healthcare workers from criminal prosecution for an unintentional medical error.

## Introduction

In March 2022, nurse RaDonda Vaught made headlines after being found guilty of two felony charges, including criminally negligent homicide, and causing serious harm-abuse or neglect of an impaired adult [1]. The conviction stemmed from a fatal medication error that occurred

**Data Availability Statement:** In accordance with ethical research standards and to protect participants' privacy, we cannot share raw interview data containing potentially identifiable information or situations that could lead to the identification of individuals involved. The interview transcripts include sensitive personal experiences, specific details, and unique situations that, if disclosed, could inadvertently expose participants' or patients' identities. Maintaining confidentiality is a cornerstone of the trust between the researcher and participants, and it is essential for ensuring honest, candid responses during the interviews. Additionally, the ethical guidelines set forth by the WVU IRB (irb@mail.wvu.edu) explicitly prohibit sharing any data that might compromise participant anonymity. Questions regarding data requests may be sent to the corresponding author, patricia.dekeseredy@hsc.wvu.edu.

**Funding:** The authors received no specific funding for this work.

**Competing interests:** The authors have declared that no competing interests exist.

during her shift at a Nashville, Tennessee hospital. Vaught promptly reported the error as required by her organization but was fired eight days later and was criminally charged. The implications of this highly publicized case may shape future nursing practice and impact the mental health and wellbeing of practicing nurses.

The reporting of mistakes in medical or nursing practices is expected of honest and ethical professionals, and organizations rely on this reporting to improve their processes and ensure patient safety. Nevertheless, primarily due to fear of punishment [2] or disciplinary action, nurses may not report medication errors that they suspect could jeopardize their employment, reputation, or nursing license [3–5]. Failing to report an error for any reason exposes the institution to risk and can threaten patient safety [6]. The Theory of Planned Behavior can help us understand whether a nurse reports an error or does not. These are 1) beliefs about the consequences of the behavior, 2) beliefs about the expectations of others, and 3) beliefs about factors that influence the performance of the behavior [7]. These beliefs influence both favorable and unfavorable behavior. For example, if a nurse views error reporting as a favorable behavior, they are more apt to report it. Conversely, if the nurse feels it is unfavorable, such as resulting in negative consequences and not meeting the expectations of others, their intention may be not to report the error. While most healthcare professionals favor reporting errors, a gap exists between reporting and actual practice [8].

Nurses who make an error can experience significant adverse psychological effects [6, 9]. Feelings of fear, shame, anxiety, guilt, and depression are common sequelae following a medical error [10]. However, less is known about the severity of these psychological symptoms and if the recent highly publicized prosecution of nurse errors can magnify the fear of reporting and influence behavior. The purpose of this study is to explore if the widespread coverage and criminal prosecution of a nurse error have influenced how nurses think about medical error reporting in a group of practicing bedside nurses.

## Methods

### Ethics statement

The West Virginia University Institutional Review Board (IRB) approved this study protocol #2204552567. The IRB approved using an informational cover letter for the consent process.

A purposive sample of nurses currently practicing in an academic tertiary care hospital was recruited from 15/05/22 to 30/07/22. Inclusion criteria included being over 18 years of age, English speaking, and a registered nurse with an unrestricted license to administer medication. Participants were recruited via snowball sampling, and a poster was placed in common areas for nurses in the hospital. Participants were emailed, or given a hard copy if in person, an IRB approved cover letter to review prior to the interview. Before the recording, the interviewer explained the study in more detail. Verbal consent was obtained after confirming understanding of the cover letter, and the participant had no further questions or concerns. Participants then completed a one-time 30 to 60-minute semi-structured telephone or in-person interview. The interviews were all conducted by one registered nurse (PD) with experience conducting qualitative interviews. There was no pre-existing relationship between the participants and the interviewer. All the interviews were digitally audio-recorded and transcribed using an artificial intelligence transcription program [11]. The interviewer checked the transcripts against the audio recordings for accuracy. The semi-structured interview guide consisted of open-ended questions generated by the three co-investigators (CS, PD, TH) asking the participants about their thoughts surrounding medical errors and the impact such a highly publicized criminal case had on their nursing practice. The analysis of the interviews was guided by conventional content analysis as described by Hsieh & Shannon [12].

Qualitative content analysis is defined as "a research method for the subjective interpretation of the content of text data through the systematic classification process of coding and identifying themes or patterns" [12 p.1278]. In conventional content analysis, the researchers avoid preconceived ideas to describe a phenomenon. First, all authors familiarized themselves with the data by reading and rereading the interview transcripts. Following this initial familiarization authors made notes of their initial impressions. Subsequently, the team of three authors developed a preliminary code book. Memo-writing and group discussion by authors CS, PD, and TH were then utilized to identify and expand themes, which were then solidified and assessed for validity by the investigators. Data analysis continued iteratively until all investigators agreed that saturation of themes had been reached. Methodological rigor was enhanced through the use of an audit trail, investigator triangulation, [13] and the use of a multidisciplinary research team. Reporting was cross-checked using the COnsolidated criteria for REporting Qualitative research (COREQ) [14].

## Findings

Participants included seven female and one male nurse. Experience ranged from 9 to over 20 years of nursing, representing seven specialties. The content analysis revealed four main themes, presented below with exemplar quotes.

## Negative disciplinary actions lead to fear of reporting medical errors

The nurses all agreed that the publicity surrounding medical errors on social media and in the news might impact the timely reporting of errors. In the Vaught case, they expressed concern that she made an error and reported it immediately as required, but her employer or leadership did not support her. Nurses felt that the cycle of events in that case, where the nurse lost her job even aside from the criminal prosecution, may make other nurses reluctant to report errors, minimize the impact of errors, or place the blame for the error elsewhere. One nurse felt there would now be a reluctance to report even minor errors.

"It's terrifying to think that, you know, it's scary that the hospital that she worked for, did not have her back." (Utilization specialist, 9 years experience)

"Something that was really concerning was, was seeing the hospital instead of standing behind that nurse who had been working for them, and working in their conditions that they've created, were allowed to sort of sit back and watch her burn. (OR nurse, 10 years experience)

Others felt the publicity of the Vaught case might lead other patients to initiate legal action against nurses, compounding the concern.

"I would definitely be more cautious about overriding anything. Not that I wasn't careful, extremely cautious before, but I think this has taken it to a different level and it changes the way I practice nursing. And also, there's a concern that people will think that anyone can just sue a nurse for anything that happens to them." (Utilization specialist- 9 years experience)

It was noted amongst nearly all nurses that Vaught suffered severe consequences despite adhering to the appropriate protocols in reporting her error. Some nurses described personal scenarios where they were disciplined or threatened with discipline over minor mistakes that they reported, which would never have been noticed if they had not self-reported. Resulting

from both personal and vicarious experiences, many nurses described current nursing practice as experiencing a breakdown of trust.

> "The people that you should be able to go to when something happens, are the people who may turn on you to protect themselves. All you have is you. Are you then going to report anything? Are you going to tell anybody? How far are you willing to go to protect yourself because you know nobody else is going to? [laughs] It's frightening to think of." (OR nurse, 10 years experience)

> "It's just disturbing to me, especially when the prosecutor [in the RaDonda case] said, "This isn't a witch-hunt in nursing." I'm like, "That's exactly what this is. This is how it's coming across. This is an absolute witch-hunt for nursing, and nobody feels safe." (OR nurse -10 years experience)

## Nurses are held responsible for system failures

The nurses discussed how systems within the hospital, which are multilevel and complex, could fail but leave them vulnerable to making a mistake and ultimately responsible for the mistake as the direct front line of patient care. For example, a double check of medication is missing, or people (such as pharmacists or MDs) are making errors at all levels of the process.

> "I think you have to like there's multiple hands that have to go through for a medication error to happen. You know, we always come back to the people that like have administered it. So it always comes back usually on nursing." (L&D nurse, 15 years experience)

> "But there's also been said that comes back on pharmacy, so there's pharmacy, there's nursing, the person who gave it there is the physician who maybe ordered it, I think you have to look at all of the aspects of everything." (L&D nurse 15 years experience)

Time limitations were interpreted as counterproductive to patient safety. One nurse questioned what the staffing may have been in Vaught's hospital on the day of the error, as nurses asked to care for an increasing number of patients limits the time they can spend on safety protocols. One nurse noted that unsafe staffing ratios led her to quit a job. She also stated that refusing such unsafe patient assignments could lead to accusations of patient abandonment or further increased workload for peers. Limitations on safety stops and "alarm fatigue" were mentioned, with several nurses noting that, eventually, the safety stops are meaningless if over-applied.

> "There's so many that it becomes it doesn't mean anything anymore. Alarm fatigue or reminders. You can only check so many times before you go crazy. Don't even read it at because you're like, we have things to do." (Neuroscience nurse, 20 years experience)

Other systems issues, such as drug shortages and substitutions resulting in different appearances of medications, were cited as possible impediments to patient safety and potentially leading to error. Overall, nurses saw themselves as vulnerable to the breakdown of such processes and at risk of being held personally responsible when system-level processes failed, or their needs or on-the-ground reality was ignored when such processes were developed.

> "We don't have the right equipment or the right supplies, but you're not looking at that, you're just telling us this is how we have to do it. [laughs]" (OR nurse, 10 years experience)

Interview participants consistently voiced concern that nursing was held to a different standard concerning hospital protocol and, in some cases, suffered comparatively more severe consequences even if their involvement was as the members of a team that should have shared responsibility:

"That was the girls that almost lost their job for the wrong site surgery. That was the thing that stuck out in my mind the most what management said they did everything right." (OR nurse, 10 years experience)

## Real life errors are common and inevitable

The participants were conscious that human error is inevitable. Most had made an error or knew someone who had. Some described near misses that could have been catastrophic for their patients. Nurses who had personally experienced an error described themselves as "walking on eggshells," ruminating upon and feeling embarrassed by their mistakes. Medication errors or near misses were commonly reported.

"There have been things when I worked in the emergency room, went to go give this kid amoxicillin. He was allergic to it." (ER nurse 13 years experience)

Often, such errors were related to verbal orders. One nurse witnessed a situation where the physician gave a verbal order for medication, and after the patient's condition worsened, the nurse felt unsupported by the ordering physician. Because of the risk of verbal orders, some nurses noted they would no longer accept them.

". . .when they had gotten a verbal order for medication and gave her medication. Well, then the physician afterwards, like the patient's blood pressure kind of tanked. So they're wondering if it was like, you know, really, like the right amount or whatever. So then that physician didn't want to say that she had told her the verbal order." (L&D nurse, 15 years experience)

Medication overrides, a specific action of Vaught, were described as something required by clinical care "everyday". Emergencies, like Vaught's scenario, were also commonly referenced as fraught with risk for errors, as systemic safeguards often delay care.

"Nurses feel it every day when someone's blood pressure drops, or they go into cardiac arrest. They're doing lifesaving mode." (CVICU 15 years experience)

## Broader negative impact of prosecuting medical errors

The nurses described how the prosecution of medical errors made them more cautious in their practice and how this would have a broader impact on the healthcare system. Verbal orders that they usually would have taken are delayed for transcription, even if this means a delay in care for the patient. A sense of self-protection was evident, and many nurses felt like they were on their own.

"I worked really hard for this and I'm not going to let somebody else destroy it for me. That's how I take my nursing career because I've just seen too many horrible things happen to good people." (OR nurse, 10 years experience)

Conversely, they felt the escalation of safeguards to prevent errors took away from their independence and use of resources.

"I would definitely be more cautious about overriding anything. And, you know, I would definitely be a little bit more careful. Not that I wasn't careful, extremely cautious before" (Utilization specialist, 9 years experience)

One nurse related that significant published medical errors relating to fatal insulin administration at another hospital led several of her patients to refuse needed insulin administration.

The participants, in general, were concerned that the current climate towards nurses would influence the nursing practice in the future. Both their personal plans for staying in clinical practice and their projections for new people entering nursing were mentioned as being influenced by the Vaught case and the general handling of nursing errors.

## Discussion

The nurses in our study described increased anxiety about making a mistake and reporting a medical error influenced by the widespread coverage of the RaDonda Vaught case. Participants recounted their own experiences with errors and the impact on their practice and discussed the broader impacts on patient care and workplace culture. The nurses felt the "weight" of the healthcare system and the enormous responsibility they bear not to make mistakes. Indeed, nurses are often called "the backbone of the hospital." providing direct patient care 24 hours a day through various interventions to improve patient outcomes, and are consistently ranked as the most trusted profession [15]. However, errors can occur while providing this care despite improved processes and advances in healthcare delivery. For nurses, errors mostly involve patient falls, infections, medication errors, documentation errors or omissions, and injuries surrounding the use of equipment [16].

Unfortunately, medical errors are expected. The World Health Organization estimates you have a 1 in 300 chance of being harmed by health care [17]. By comparison, air travel has a 1 in a million chance of harm [17]. In an extensive study of over 10,000 ambulatory care patients, 36% felt they were involved in a medical error [18]. It is estimated that the overall occurrence of adverse events for hospitalized patients is 10%, with half of those considered preventable [19]. Medical errors are distressing for all parties involved, including patients, families, and healthcare professionals. Participants in our study who shared their personal experiences with medical errors described the impact on their mental health, such as feeling fearful and embarrassed and how these mistakes can shape future nursing practice. System failures at all levels of healthcare delivery can leave nurses susceptible to errors [6]. Our participants felt breakdowns in communication made it much less likely they would take a verbal order from a physician. Instead, care is delayed until the order is written in the chart and processed through the system. This defensive behavior was to reduce the risk of making a mistake. Defensive measures by nurses impact other aspects of patient care, such as keeping excessively detailed notes, avoiding litigious patients, and questioning the need for certain medications [20].

The case against a nurse often makes news when fatal errors occur, and the healthcare worker's actions face intense scrutiny. In the Vaught case, even though the proper protocols for reporting and investigating the mistake were followed, the nurse faced severe consequences. These actions can have devastating results for the mental health of the nurse involved. In 2011, a 50-year-old nurse, Kimberly Hiatt, committed suicide seven months after a tragic drug miscalculation led to the death of a critically ill child [21]. Ms. Hiatt, from all reports, was an exemplary nurse who loved her job. She had no history of violating safety protocols, yet she

made a mistake [22]. This case was also highly publicized at the time. In a Washington State Nurses Association survey shortly after this incident, half of the nurses felt an error would be held against them [23]. In the same study, close to a third reported hesitancy in reporting errors because of punitive measures and reactions from higher management [23]. The nurses in our study shared similar concerns and hesitancy with reporting errors related to the lack of support and anticipated retribution against the one who made a mistake. Fear of consequences is the number one reason for underreporting medical errors [24]. When these consequences play out in the media, such as the criminal prosecution of health care providers, it heightens this fear and can impact patient care [25].

The demands and working conditions associated with nursing are known to impact the mental health of these professionals and can be especially magnified in smaller hospitals with limited resources [26]. Compounded by the ease and availability of social media and speedy internet access, information regarding the public prosecution of medical errors spreads quickly and can produce strong emotions within this group of workers. Emotions ranged from disbelief that such an egregious error could occur to outrage at her criminal prosecution and license removal. The proliferation of online discussions about the RaDonda Vaught case prompted the American Nurses Association (ANA) to release a statement of concern suggesting that criminalizing medical errors could have a chilling effect on reporting and process improvement [27]. Despite this alert, few studies have been conducted following the conviction to quantify these concerns. The nurses in our study confirmed the concerns of the ANA and were negatively influenced by the media coverage of the criminal case against RaDonda Vaught.

In 2000, a seminal book was written about the prevalence of medical errors and cultivating an environment of safety to encourage reporting of errors [28]. This book shifted the focus of medical errors away from the individual and called for systemic change to improve patient safety [28]. A 2019 documentary, To Err is Human, described how medical professionals who fear the consequences of reporting may not disclose the mistake, lie about the error, or try to cover it up [29]. However, even with multiple levels of checks and balances throughout the hospital system, mistakes will be inevitable. Therefore, it becomes essential to be mindful of the optics associated with how those who report errors are treated within their institutions and in society. If reporting errors is perceived negatively amongst peers in the institutional or broader environment, then they will be less likely to disclose errors [7]. The criminal prosecution of RaDonda Vaught, along with her employer's perceived lack of support, left many nurses thinking, "This could be me".

There have been efforts to respond to the RaDonda case to ease the concerns voiced by nurses and other healthcare providers that a mistake at work could leave them with a criminal record and in professional ruin. In March 2024, Kentucky passed House Bill 159 to provide healthcare workers immunity from criminal liability for harm caused by a health-related act or omission [30]. There are exceptions for willful, malicious, or intentional damage. Kentucky is the first state to enact protections for its healthcare workers from criminal prosecution of medical errors. The decriminalization of medical errors in Kentucky is a first step in addressing the fear and mental health implications of reporting mistakes and facilitates communication to improve the delivery of compassionate, quality care.

There have been other efforts to support the reporting of errors. The concept of just culture in healthcare shifts the focus from individual blame and subsequent punitive actions to system design behavioral management [31]. Just culture aims to encourage the open and honest reporting of errors. However, challenges remain when considering the impact of emotions and privacy for those involved [32]. The open communication underpinning just culture becomes problematic when an error is broadly publicized and an individual is named the

guilty party. Incorporating a balance between openness and accountability with emotions, facts, and learning from incidents is suggested [32]. Future research, including supportive mental health interventions, legal protections for nurses, and expanding the study to include management and other healthcare professionals, is indicated.

There are some limitations to this work. First is the potential for bias in recall or reporting, particularly in discussing personal nursing errors. Some participants may have omitted severe errors they observed or participated in because of the nature of such disclosures. We attempted to mitigate this risk by conducting interviews with a peer (RN). Nurses who volunteered for this study may also be ones who are more open to discussing errors, and those who are more uncomfortable and less apt to report may not be represented. Our participants are from one hospital in an extensive statewide healthcare system. Findings may not be generalizable to small or rural hospitals or to nurses working in other states. Additionally, our sample size of eight nurses may be considered low; however, we mitigated this by reporting on themes with particular salience across all participants.

## Conclusions

A highly publicized case in which a nurse has lost her job and is found criminally responsible for a reported medical error can have a lasting impact on the nursing profession. Participants in this study confirmed that anticipated negative disciplinary actions lead to fear of reporting medical errors; however, they know that real-life mistakes are common and inevitable. Moreover, nurses understand and are concerned that they are one piece of a more extensive system yet can be held responsible for system failures. The criminal prosecution and ensuing media coverage of nurse errors are believed to have a broader negative impact on healthcare, patient safety, and workplace wellbeing. Despite these concerns, more work is indicated to identify safe and constructive ways to handle medical errors that do not perpetuate fear of reporting them. As in Kentucky, decriminalization and immunity from criminal prosecution for healthcare workers could be an important part of this process.

## Author Contributions

**Conceptualization:** Patricia Dekeseredy, Cara L. Sedney.

**Investigation:** Patricia Dekeseredy, Cara L. Sedney.

**Methodology:** Patricia Dekeseredy, Cara L. Sedney.

**Project administration:** Patricia Dekeseredy.

**Software:** Patricia Dekeseredy.

**Writing – original draft:** Patricia Dekeseredy, Treah Haggerty, Cara L. Sedney.

**Writing – review & editing:** Patricia Dekeseredy, Treah Haggerty, Cara L. Sedney.

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
