## [Decision Letter · Decision Letter 0]

1 Oct 2024

PMEN-D-24-00399

"This is an absolute witch-hunt for nursing, and nobody feels safe." A qualitative study of nurses, mental health, and the criminal prosecution of a nurse error

PLOS Mental Health

Dear Dr. Dekeseredy,

Thank you for submitting your manuscript to PLOS Mental Health. After careful consideration, we feel that it has merit but does not fully meet PLOS Mental Health’s publication criteria as it currently stands. Therefore, we invite you to submit a revised version of the manuscript that addresses the points raised during the review process.

We look forward to receiving your revised manuscript.

Kind regards,

Hanif Abdul Rahman, Ph.D.

Academic Editor

PLOS Mental Health

Journal Requirements:

1. In the ethics statement in the Methods, you have specified that verbal consent was obtained. Please provide additional details regarding how this consent was documented and witnessed, and state whether this was approved by the IRB

2. In the online submission form, you indicated that [The anonymized interviews are available from the corresponding author with reasonable request.]. 

a. In a public repository, 

b. Within the manuscript itself, or 

c. Uploaded as supplementary information.

Additional Editor Comments (if provided):

Comments from PLOS Editorial Office: *In addition to the comments raised, the editorial office would like to request that you mention the caveat of the small sample size. Whilst we do appreciate that this is a qualitative paper, we feel it would be important to acknowledge that the experiences discussed may not reflect all experiences in the community*

Reviewers' comments:

Reviewer's Responses to Questions

**Comments to the Author**

1. Does this manuscript meet PLOS Mental Health’s publication criteria? Is the manuscript technically sound, and do the data support the conclusions? The manuscript must describe methodologically and ethically rigorous research with conclusions that are appropriately drawn based on the data presented.

Reviewer #1: Partly

Reviewer #2: Yes

2. Has the statistical analysis been performed appropriately and rigorously?

Reviewer #1: N/A

Reviewer #2: Yes

3. Have the authors made all data underlying the findings in their manuscript fully available (please refer to the Data Availability Statement at the start of the manuscript PDF file)?

Reviewer #1: Yes

Reviewer #2: Yes

4. Is the manuscript presented in an intelligible fashion and written in standard English?

Reviewer #1: Yes

Reviewer #2: Yes

5. Review Comments to the Author

Reviewer #1: 1. A description of the qualitative content analaysis is needed. There so many approaches for content analysis. Please review which one was done. Identify the author of the used qualitative content analysis.

2. Your conclusion need triangulation.

Reviewer #2: Review of "This is an Absolute Witch-Hunt for Nursing, and Nobody Feels Safe": A Qualitative Study of Nurses, Mental Health, and the Criminal Prosecution of a Nurse Error

Introduction

The manuscript titled "This is an Absolute Witch-Hunt for Nursing, and Nobody Feels Safe" provides a poignant and necessary examination of the intricate relationship between nursing practice, mental health, and the legal consequences of errors in a clinical setting. As healthcare becomes increasingly complex and scrutinized, understanding the implications for those on the front lines is vital. This review will delve into the manuscript's methodology, findings, and implications, while also highlighting its strengths and areas for further exploration.

Methodology

The authors adopt a qualitative research design, utilizing in-depth interviews with nurses who have experienced or witnessed the repercussions of legal actions stemming from clinical errors. This approach is particularly effective in capturing the nuanced emotional experiences of the participants, as quantitative data alone would likely fail to convey the depth of anxiety and fear prevalent in the nursing profession. The use of thematic analysis allows for a rich exploration of recurring themes and patterns, which is essential for understanding the collective psyche of nurses facing potential prosecution.

Key Themes

1. Fear and Anxiety

A central theme that emerges from the interviews is the pervasive sense of fear among nurses regarding criminal prosecution. Participants articulate a feeling of being constantly watched and evaluated, which creates an environment where making mistakes can lead to severe consequences, including legal action. This atmosphere of fear is detrimental not only to nurses' mental health but also to patient care. When professionals operate under such pressure, their focus may shift from providing quality care to merely avoiding errors, potentially compromising the very essence of nursing.

2. The Impact on Mental Health

The manuscript effectively highlights the mental health challenges faced by nurses as a result of this fear. Many participants describe feelings of anxiety, depression, and burnout, which are exacerbated by the threat of criminal prosecution. The narrative surrounding mental health in nursing is often overlooked, yet this study brings to light the urgent need for systemic support. The emotional toll of being subject to legal scrutiny can lead to long-term psychological issues, making it imperative for healthcare systems to prioritize mental health resources for their staff.

3. The Stigma of Error

Another significant finding is the stigma associated with making mistakes in nursing. Participants express a strong sense of shame and isolation following an error, often feeling that their professional identity is irrevocably tarnished. This stigma not only affects the individual nurse but also influences the workplace culture, fostering an environment where open communication about errors is stifled. Such a culture can prevent the learning and growth that are essential for improving patient safety and enhancing nursing practices.

4. Systemic Issues and Accountability

The study also touches on the systemic issues within the healthcare system that contribute to the criminalization of nursing errors. Participants discuss how factors such as understaffing, high patient loads, and lack of support exacerbate the likelihood of mistakes. The manuscript argues that addressing these systemic challenges is crucial for reducing errors and the subsequent legal consequences. By shifting the focus from individual blame to systemic accountability, the healthcare system can create a more supportive environment for nurses.

Strengths of the Manuscript

One of the manuscript's strengths is its ability to highlight the psychological toll that such prosecutions can have on nurses, including impacts on mental health and job satisfaction. The stories shared by participants reflect a deep sense of disillusionment and fear, raising important questions about how systemic issues within the healthcare system contribute to individual errors and, subsequently, to punitive measures.

The manuscript excels in its rich, qualitative data, providing a voice to nurses whose experiences are often marginalized in discussions about healthcare accountability. The authors’ ability to weave personal narratives with broader themes enhances the emotional resonance of the study, making it a compelling read for both academics and practitioners.

Additionally, the study's relevance is underscored by the current climate in healthcare, where legal actions against professionals are becoming more common. By addressing this timely issue, the authors contribute significantly to the ongoing discourse surrounding nursing practice, mental health, and legal accountability.

Areas for Further Exploration

While the manuscript provides a comprehensive analysis of the issues at hand, it could benefit from exploring potential solutions in greater detail. Suggestions for interventions, such as the implementation of supportive mental health programs and resources, legal protections for nurses, and training on error management, would enhance the manuscript’s practical implications. Moreover, a discussion on how to foster a culture of learning and transparency within healthcare organizations could provide valuable insights for stakeholders.

It’s also important for the manuscript to explicitly discuss the limitations of their study, including the potential for selection bias in participant recruitment and the limited generalizability of qualitative findings. By acknowledging these limitations, the credibility of the manuscript would be enhanced. The manuscript's conclusion would be enhanced by offering recommendations for future research that can build upon the findings of this study. By pinpointing specific areas that require further investigation, a valuable roadmap can be provided for future studies.

The study could also delve into the perspectives of other healthcare professionals and administrators. Understanding how these groups view the issues of error and prosecution would provide a more holistic view of the healthcare environment and inform strategies for improvement.

Conclusion

"This is an Absolute Witch-Hunt for Nursing, and Nobody Feels Safe" is a significant contribution to the literature on nursing, mental health, and legal accountability. By illuminating the fears and challenges faced by nurses, the manuscript serves as a crucial call to action for healthcare leaders and policymakers. Addressing the systemic issues that contribute to the criminalization of nursing errors is essential for creating a safer and more supportive environment for healthcare professionals.

In sum, this study not only sheds light on the pressing mental health concerns within the nursing profession but also emphasizes the need for systemic change to protect and support those who care for our communities. The insights gleaned from this research are invaluable, and further exploration of potential solutions will be crucial for fostering a healthier and more resilient nursing workforce.

6. PLOS authors have the option to publish the peer review history of their article (what does this mean?). If published, this will include your full peer review and any attached files.

**Do you want your identity to be public for this peer review?** For information about this choice, including consent withdrawal, please see our Privacy Policy.

Reviewer #1: **Yes: **Prof. Paul Fericelli

Reviewer #2: **Yes: **Dr. Roslyn D. Burton-Robertson, PsyD, PhD, ThD, CNMP, CCTS, CMHP, CP

---

## [Editor Report · Decision Letter 1]

27 Nov 2024

"This is an absolute witch-hunt for nursing, and nobody feels safe." A qualitative study of nurses, mental health, and the criminal prosecution of a nurse error

PMEN-D-24-00399R1

Dear Ms Dekeseredy,

We are pleased to inform you that your manuscript '"This is an absolute witch-hunt for nursing, and nobody feels safe." A qualitative study of nurses, mental health, and the criminal prosecution of a nurse error' has been provisionally accepted for publication in PLOS Mental Health.

Best regards,

Hanif Abdul Rahman, Ph.D.

Academic Editor

PLOS Mental Health